# Identification and Functional Analysis of the Cell Proliferation Regulator, Insulin-like Growth Factor 1 (IGF1) in Freshwater Pearl Mussel (*Hyriopsis cumingii*)

**DOI:** 10.3390/biology11091369

**Published:** 2022-09-19

**Authors:** Xuenan Li, Shangle Feng, He Wang, Xiaoya Shen, Yige Chen, Xingrong Xuan, Yuanshuai Fu, Zhiyi Bai, Wenjuan Li

**Affiliations:** 1Key Laboratory of Freshwater Aquatic Genetic Resources, Ministry of Agriculture and Rural Affairs, Shanghai Ocean University, Shanghai 201306, China; a15165221580@163.com (X.L.); fengshangle@126.com (S.F.); wh726444887@163.com (H.W.); shenxiaoya224@163.com (X.S.); cyg8910cyg@163.com (Y.C.); xrxuan@shou.edu.cn (X.X.); ysfu@shou.edu.cn (Y.F.); 2Shanghai Collaborative Innovation for Aquatic Animal Genetics and Breeding, Shanghai 201306, China; 3Shanghai Engineering Research Center of Aquaculture, Shanghai Ocean University, Shanghai 201306, China

**Keywords:** *Hyriopsis cumingii*, *HcIGF1*, RNA interference, proliferation activity

## Abstract

**Simple Summary:**

*Hyriopsis cumingii* (*H. cumingii*, mussels) is an important freshwater pearl shellfish species, and its mantle is the most important tissue during shell and pearl mineralization. However, the low proliferative capacity of mantle cells during in vitro culture hinders the establishment of stable cell lines. In this study, we identified and explored the function of IGF1. The results show that IGF1 has a typical IIGF structural domain, exerts biological functions through the PI3K/AKT signaling pathway, and that the recombinant protein significantly promotes the proliferation of mantle cells. Our findings will help to further extend the culture time of mantle cells as well as to understand the growth regulation mechanism of mussels.

**Abstract:**

Insulin-like growth factor 1 (IGF1) plays an important regulatory role in the regulation of growth, differentiation, and anabolism in a variety of cells. In this study, the full-length cDNA of the *IGF1* gene was cloned from *Hyriopsis cumingii*, named *HcIGF1*. The expression level of *HcIGF1* in six tissues (adductor muscle, foot, hepatopancreas, gill, mantle, and gonad) was determined. In addition, the localization of *HcIGF1* in the mantle was analyzed by in situ hybridization, and finally the function of *HcIGF1* was explored by RNA interference and prokaryotic expression. The results showed that the amino acid sequence contained a typical IIGF structural domain. The phylogenetic tree showed that HcIGF1 clustered with other marine bivalve sequences. Quantitative real-time PCR and in situ hybridization analysis showed that *HcIGF1* was expressed in all tissues. The highest expression was in the foot and the lowest was in the mantle. In the mantle tissue, the hybridization signal was mainly concentrated in the outer mantle. After RNA interference, the expression of *IGF1* was found to be significantly decreased (*p* < 0.05), and its related genes *IGF1R*, *AKT1*, and *cyclin D2* were downregulated, while *MAPK1* were upregulated. The recombinant HcIGF1 protein was purified and its growth-promoting effect was investigated. The results showed that the recombinant HcIGF1 protein could significantly promote the proliferative activity of the mantle cells of mussels, with the best proliferative effect at 12.5 μg/mL. The results of this study provide a new method to solve the problem of weak proliferation of shellfish cells in vitro and lay the foundation for further understanding of the growth regulation mechanism of *H. cumingii*, as well as a better understanding of the physiological function of IGF1 in mollusks.

## 1. Introduction

Insulin-like growth factors are a class of peptides that are similar in metabolic function and structure to insulin [1]. Insulin-like growth factor 1 (IGF1) is a member of the insulin-like growth factor family, and consists of four functional regions, B-C-A-D. It plays a key role in vertebrate growth and development, and is involved in regulatory processes such as cell proliferation, differentiation, migration, and apoptosis [2,3,4].

IGF1 has been shown to have diverse physiological functions in different tissues, involving growth, development, metabolism, mineralization, and other physiological regulatory processes [5,6,7]. In mammals, IGF1 research has focused on tumors and growth disorders. It has been found that overexpression of *IGF1* may increase the probability of overgrowth, growth flocculation and tumor formation [8], and growth retardation after being knocked out [9]. It has been shown that IGF1 has a significant pro-proliferative effect. In vitro treatment with IGF1 can reintroduce normal fibroblasts cultured by serum starvation method into S-phase. Furthermore, the addition of IGF1 increased the expression level of cyclin D1 protein, which accelerated the G1 phase of the cell cycle and promoted the proliferation of rumen epithelial cells [10]. Among aquatic organisms, *IGF1* has been most studied in fish. Since the first cloning and identification of IGF1 in *Oncorhynchus kisutch* [11], *IGF1* has been cloned in *Lates calcarifer* [12], *Morone saxatilis* [13], *Sparus aurata* [14], *Danio rerio* [15], and *Carassius auratus* [16]. Studies have also shown the important physiological functions of IGF1 in fish growth and development. The exploration of the physiological functions of IGF1 has led to increased research on IGF1; however, IGF1 has not yet been reported in mollusks.

Secreted IGF1 needs to be bound by IGF binding proteins to reach the target cells, where they bind to insulin-like growth factor receptor 1 (IGF1R) on the surface of the target cells and cause autophosphorylation [17]. These molecules then interact with downstream sensing and effector molecules, opening two major intracellular signaling pathways: the mitogen-activated protein kinase (MAPK) pathway and the phosphatidylinositol 3 kinase/protein kinase B (PI3K/AKT) pathway, thus exerting mitogenic, metabolic, and antiapoptotic biological functions [18,19]. It has been shown that activation of MAPK and PI3K/AKT is the downstream mechanism that mediates the IGF1 signaling cascade to promote differentiation of spermatogonia to primary spermatocytes (Figure 1). A large number of experiments have shown that IGFs can prevent apoptosis through the same signaling pathway in different cell types. In the same cell, IGFs can function through multiple signaling pathways. Different signaling pathways are also not independent of each other, but are interconnected and interact with each other, showing a complex reticulation regulatory structure [20,21].

*Hyriopsis cumingii* is an important freshwater pearl mussel species, accounting for more than 70% of global freshwater pearl production [22]. The mantle is the most important tissue in shell and pearl mineralization [23], but its low proliferative capacity during in vitro culture hinders the establishment of stable cell lines [24]. In addition, little research has been conducted on the growth regulation mechanism of *H. cumingii*, which has hindered the development of the industry to some extent. As one of the essential factors regulating growth and development, IGF1 has important research value. In this study, the full-length cDNA of *HcIGF1* was cloned, and the function of HcIGF1 was investigated by in situ hybridization, RNA interference (RNAi), and prokaryotic expression. It provides a theoretical basis for addressing the weak proliferation ability of set cells in vitro and prolonging culture time, as well as laying a foundation for further understanding the growth regulation mechanism of *H. cumingii* and the physiological function of IGF1 in mollusks. This has long-term significance for improving the growth performance of mussels, accelerating the breeding process and developing the freshwater pearl industry.

## 2. Materials and Methods

### 2.1. Experimental Animals and Tissue Samples

Fifty healthy mussels (~1 year old, 5–6 cm) were purchased from Wuyi farm (Wuyi, China). The mussels were all housed in a 120 L aquarium with aerated tap water for one week. Chlorella was fed twice daily and mixed well. The water temperature was kept at 25 ± 2 °C using a thermostatic thermometer. 

Tissues including gill, mantle, gonad, hepatopancreas, adductor muscle, and foot were dissected from the mussels with autoclaved scissors, forceps, and scalpel. Tissue samples were placed in liquid nitrogen for rapid freezing and stored at −80 °C prior to RNA extraction.

### 2.2. RNA Isolation and cDNA Synthesis

Samples were removed from the −80 °C refrigerator and placed on ice. Enzyme-free steel beads were then added and placed on the homogenizer for homogenization. Total RNA was extracted from each tissue sample using TRIzol (TaKaRa, Shiga, Japan) and RNA concentration and purity were determined by Nano Drop 2000 C (Thermo Scientific, Waltham, MA, USA). A 2 μL sample of RNA solution was taken to check the integrity of the RNA by 1% agarose gel electrophoresis. Evo M-MLV Reverse Transcription Premix Kit (Accurate Biology, Changsha, China) was used to reverse-transcribe the RNA into cDNA.

### 2.3. Cloning of Full-Length cDNA

The first strand of cDNA was synthesized by the SMART 5′ RACE and 3′ RACE kit (Clontech, San Francisco, CA, USA) and the 3′ and 5′ ends were amplified according to the kit instructions. Primers were designed using Primer 5.0 based on *IGF1* sequences screened from the mussel mantle transcriptome library and synthesized by Sangon Biotech (Shanghai, China) (Table 1). The PCR products were purified according to the SteadyPure Agarose Gel DNA Purification Kit (Accurate Biology, Changsha, China). The recovered purified product was ligated onto pCEM-T Easy (Promega, Madison, WI, USA) for 5 h at 16 °C. Ligation system: 1 μL of PMD19-T Vector, 5 μL of Solution I, 4 μL of recovered product. Then, 10 μL of ligated product was transferred into 100 μL of DH5α. After 90 s of heat excitation, 1 mL of LB liquid medium was added and incubated on a shaker for 2 h. The plates were coated and incubated for 10 h~12 h at 37 °C. Individual colonies were picked and incubated in LB liquid medium, and six positive clones were randomly selected and sent to Sangon Biotech (Shanghai, China) for sequencing. The sequences obtained from sequencing were spliced to obtain the full length of the target gene *IGF1* sequence, named *HcIGF1*.

### 2.4. Bioinformatics Analysis

BLAST from the NCBI database (https://blast.ncbi.nlm.nih.gov/Blast.cgi, accessed on 15 March 2022) was used to perform sequence similarity analysis of IGF1. ORF finder (https://www.ncbi.nlm.nih.gov/orffinder/, accessed on 15 March 2022) was used to determine amino acid sequences and identify open reading frames. ProtParam (https://web.expasy.org/protparam/, accessed on 15 March 2022) was used to analyze physical parameters. SMART (http://smart.embl-heidelberg.de/, accessed on 15 March 2022) and NCBI databases were used to predict structural domains. GeneDoc2.7 was used for amino acid sequence alignment. TMHMM Server v2.0 (http://www.cbs.dtu.dk/services/SignalP/, accessed on 15 March 2022) was used to predict transmembrane structures and Phyre2 (http://www.sbg.bio.ic.ac.uk/phyre2/html/page.cgi?id=index, accessed on 15 March 2022) was used to predict the secondary structure. The phylogenetic tree was constructed using the Clustal W2 program and the neighbor-joining (NJ) method in MEGA7.0 (Mega Limited, Auckland, New Zealand). The main parameters were as follows: test of phylogeny—bootstrap method; No. of bootstrap replications—1000; method—p-distance; rates among sites—uniform rates; site coverage cutoff (%)—50. Finally the Jukes–Cantor one-parameter model was used to calculate the degree of divergence of the sequences.

### 2.5. Quantitative Real-Time PCR (qRT-PCR) Analysis

RNA extraction and cDNA synthesis methods were described in Section 2.2. QRT-PCR-specific primers were designed using NCBI for the ORF region of the *HcIGF1* gene, with *EF1α* as the internal reference gene (GenBank accession number GW694601) (Table 1). Standard curves for each primer were generated using a 10-fold serial dilution of cDNA as template, based on a linear regression model. The amplification was performed in triplicate on Bio-Rad CFX96 (Bio-Rad, Hercules, CA, USA) using SYBR^®^ Green Premix Pro Taq HS qPCR Kit (Accurate Biology, Changsha, China). Each reaction was performed in a 20 µL volume containing 10 µL of 2× SYBR^®^ Green Pro Taq HS Premix II, 0.8 µL of forward primer (10 μmol/L), 0.8 µL of reverse primer (10 μmol/L), 6.8 µL of ddH_2_O, and 1.6 µL of cDNA (equivalent to 100 ng of total RNA). Non-template controls were also included for each primer pair. The amplification conditions were 95 °C for 3 min, 95 °C for 5 s, and 60 °C for 30 s; 40 cycles. Signals were collected during the extension phase, and each cycle increased 0.5 °C for 5 s from 65 °C to 95 °C, followed by collecting the fluorescence signal of the dissolution curve. The qRT-PCR analyses were completed with three biological replicates and three technical replicates. The amplification results were analyzed by the 2^−ΔΔCT^ method to obtain the expression of each sample relative to the internal reference gene EF1α [25]. 

### 2.6. In Situ Hybridization

The specific primers were designed with Primer 5.0 (Table 1), and the purified PCR products were used as in vitro transcription templates to obtain labeled probes using the T7 High Efficiency Transcription kit (TransGen Biotech, Beijing, China) and DIG RNA Labeling Mix (Roche, Germany), and the probes were purified.

The mantle tissues of the mussels were fixed in 4% paraformaldehyde for 24 h at 4 °C and stored in 70% ethanol (prepared DEPC water). The tissues were gradient-dehydrated, embedded, and sectioned with a paraffin slicer (Leica, Heidelberg, Germany) at a thickness of 4 μm. Then, in situ hybridization was performed according to the instructions of the DIG nucleic acid detection kit (SP6/T7; Roche, Germany), treated with DAB chromogen for light avoidance, and the hybridization signal was observed and photographed using a Leica DM 2500 microscope (Leica, Heidelberg, Germany).

### 2.7. Synthesis of dsRNA In Vitro and In Vivo Interference

Based on the DNA as a template, the T7 promoter was spliced upstream of the sense and antisense strands of the target sequence. After using polymerase, each was transcribed in vitro to obtain two single-stranded RNAs. Double-stranded RNA (dsRNA) was formed after annealing, and the interference strand of the desired dsRNA was obtained after purification. Three pairs of RNAi-specific primers were designed according to the ORF sequence by primer 5.0 (Table 1). The method of dsRNA interference strand synthesis of *HcIGF1* gene is referred to Wang et al. [26]. The three interfering chains synthesized were named as interfering chain 1 (G1), interfering chain 2 (G2), and interfering chain 3 (G3).

A negative control group (injected of RNase/DNase-free water) and three experimental groups (injection of interfering chains) were designed in this experiment, with 15 mussels in each group. Each animal was injected with 100 μL of interference chain into the adductor muscle with a 1 mL disposable syringe. To determine the timing of gene silencing, three time points (24, 48, and 72 h) were established. In the negative control group and the experimental group, 5 mussels were randomly selected at each time point after injection to obtain mantle tissue, and then RNA extraction and cDNA synthesis were performed for subsequent gene expression detection. The gene sequences used to design the primers were all obtained from the transcriptomic library of the mussel obtained from previous experiments, of which the *cyclin D2* gene has been uploaded to NCBI (GenBank accession number MZ558548). Refer to Section 2.2 for specific RNA extraction and cDNA synthesis methods.

### 2.8. Prokaryotic Expression and Activity Analysis of HcIGF1

Based on the cloned HcIGF1 ORF sequence, we designed primers and added an *EcoR*I site to amplify the target fragment with the enzyme cut site (Table 1). The PCR amplification conditions were: 94 °C for 3 min; 94 °C for 30 s, 50 °C for 30 s, 72 °C for 30 s, 35 cycles; 72 °C extension for 5 min. For the method used to recover the target fragment, refer to Section 2.3. The pET32a prokaryotic expression vector was digested using *EcoR*I for 1 h at 37 °C. The target fragment was ligated to the vector following the Seamless Cloning Kit (Beyotime, Shanghai, China) instructions. Incubation of the recombinant expression vector with BL21 receptor cells was conducted at 37 °C in LB liquid medium. Once the OD600 reached around 0.6, IPTG was added at a final concentration of 1 mmol/L. Three sets of experiments were designed: 3 h induction at 25 °C, 3 h induction at 37 °C, and 5 h induction at 15 °C. The cells were collected by centrifugation and crushed by ultrasound. The protein was purified using the His-tag Protein Purification Kit (Beyotime, Shanghai, China), and the concentration of the purified recombinant protein was determined using the BCA kit (Beyotime, Shanghai, China). 

The recombinant proteins were electrophoresed by SDS-PAGE and transferred to Trans-Blot SD Semi-Dry Electrophoretic Transfer Cell (Bio-Rad, Hercules, CA, USA) to enable protein transfer to the PVDF membrane following the method described in Han et al. [27]. Then, closure solution was used for 2 h. After three washes with TTBS, 1:1000 His-tag antibody was added (mouse monoclonal antibody, Thermo Scientific, Waltham, MA, USA) and incubated for 10 h. After three washes, 1:1000 diluted secondary antibody was added (Thermo Scientific, Waltham, MA, USA) and incubation continued for 3 h. After three washes, lasting 8 min each, the ECL chemiluminescence hypersensitive color development kit (Beyotime, Shanghai, China) was used to develop color, and ChemiDoc XRS + (Bio-Rad, Hercules, CA, USA) was used for photography.

Ten 1-year-old mussels of uniform size were selected and non-anticoagulated blood was drawn from the adductor muscle area with a sterile 1 mL syringe, placed in corresponding 15 mL centrifuge tubes, centrifuged for 20 min at 3000× *g*, filtered through 0.45 μm and 0.22 μm membranes, and sealed and stored in a −20 °C refrigerator.

The 1-year-old mussels were taken from the laboratory and sprayed with 75% alcohol and left to rest for 20~30 min. The adductor muscle was cut with a scalpel. After opening the shells, the mantle area on both sides of the mussels was wiped with cotton balls and the mantle tissue was removed. The tissue was washed once in sterile centrifuge tubes with PBS buffer and then repeatedly washed 3 to 5 times with a final concentration of 2% triple antibiotics (penicillin, streptomycin, and amphotericin B; Beyotime, Shanghai, China) configured in PBS buffer. The mantle tissue was cut to about 1 mm^2^ in size with autoclaved scissors and digested by adding 10 mL of 0.25% trypsin-EDTA. The mantle tissue was digested overnight at 4 °C for 12 h. At the end of digestion, it was heat-digested at 37 °C for 20 min. The digestion was terminated by adding the same volume of medium containing 10% autologous serum and left at room temperature for 10 min. The cell suspension was obtained by filtering the mantle tissue with 200-mesh and 400-mesh filters, respectively.

The mussel mantle cells were inoculated into 96-well plates with 1 × 10^5^ cells per well, and 100 μL of culture medium (10% mussel autologous serum + RPMI 1640 basal medium + 1% triple antibiotics) was added to each well and incubated at 25 °C in a constant temperature incubator without CO_2_. After the cells adhered to the wall, the medium was removed and incubation continued with a medium containing different concentrations of purified protein (0, 2.5, 5, 7.5, 12.5, 17.5, and 22.5 µg/mL; each group had six replicates) and 10 μL CCK8 solution was added to each group at 0, 1, 2, 3, 4, 5, and 6 days, respectively. The incubation lasted for 2 h. The OD values at 450 nm were measured.

### 2.9. Statistical Analysis

Gene expression levels were calculated as the number of cycles needed for the amplification to reach a fixed threshold in the exponential phase of the PCR (i.e., Ct). The threshold was set to 500 for all genes. All data were statistically analyzed and plotted using SPSS 22.0 and Origin 9.1. One-way analysis of variance (ANOVA) was used to analyze differences in gene expression and differences in cell viability after treatment with different concentrations of recombinant protein based on the Ct and OD results obtained from the experiments. Experimental values were expressed as the mean ± SE. Multiple comparisons were analyzed using Tukey’s method and *p* < 0.05 was considered statistically significant.

## 3. Results

### 3.1. Bioinformatics Analysis of HcIGF1

In this experiment, the full-length cDNA of the *HcIGF1* gene was obtained by RACE cloning at 1507 bp, of which the 3′UTR is 434 bp, the 5′UTR is 365 bp, and the ORF is 708 bp, with 235 predicted amino acids, and the amino acid sequence of this gene has an IIGF structural domain located at residues 55–1126 (Figure 2). The molecular weight of the protein was 26.27 kDa and the theoretical isoelectric point was 9.21, with serine being the most abundant amino acid, accounting for 11.1% of the total amino acid content. There was no obvious transmembrane region. The GenBank accession number is OM962969. 

The amino acid sequences of nine representative species of IGF1 were selected from the NCBI database for homology alignment analysis. The results showed 80.43% similarity to the IGF1 sequence of *Potamilus streckersoni* and 77.87% similarity to the IGF1 sequence of *Amblema plicata* (Figure 3).

The results of the phylogenetic tree indicate that the IGF1 protein of *H. cumingii* clustered with *A. plicata*, *P. streckersoni*, and other bivalves (Figure 4). The molecular evolutionary position of *HcIGF1* in the phylogenetic tree is consistent with the taxonomic position of the mussels.

### 3.2. HcIGF1 Expression in Different Tissues

The PCR specificity of all genes showed a peak by analysis of the melting curves. The amplification efficiencies for all genes were 91.0~109.4%, indicating that the selected quantitative primer pairs were well-designed and had good amplification efficiency and specificity. The non-template control results showed that all primer pairs did not show amplification, or did not show amplification until after 36 Ct. The qRT-PCR results revealed that *HcIGF1* was expressed in all tissues, and the expression level was higher in foot, adductor muscle, and gonad, with the highest expression level found in the foot. The expression level was low in gill, mantle, and hepatopancreas, with expression in the mantle being the lowest (Figure 5, Appendix A).

### 3.3. In Situ Hybridization

In order to investigate the expression of *HcIGF1* in the mantle, we localized the *HcIGF1* gene in the mantle of mussels. The results of in situ hybridization showed that in the experimental group, there were obvious brown-yellow hybridization signals in the mantle, mainly located in the outer fold, ventral mantle, and middle fold of the mantle, with the strongest hybridization signals in the ventral mantle. No signal was detected in the control group (Figure 6, Appendix A).

### 3.4. RNAi and Effects on Downstream Genes

After interference, the relative expression of the *HcIGF1* gene was detected by qRT-PCR. The results showed that all three interfering chains could effectively reduce the expression level of *HcIGF1* (*p* < 0.05), although the G2 and G3 interfering chains had stronger inhibitory effects. Compared with the negative control group, the G2 interfering chain reduced gene expression by 89.2% and 72.9% at 48 h and 72 h after injection, respectively. The G3 interfering chain showed a significant interfering effect 24 h after injection. The interference rates were 64.24%, 90.8%, and 75.7% at 24, 48, and 72 h, respectively, compared to the negative control group. Therefore, among the three synthetic interfering chains, G3 had the best interfering effect (Figure 7A; Appendix A).

In order to study the effect of related signaling pathways after interference, four downstream genes, *IGF1R*, *cyclin D2*, *AKT1,* and *MAPK1*, were selected for qRT-PCR to detect the relative expression levels of the genes. In this experiment, the mantle tissues after interference with the G3 interference chain was selected for the next part of the study. The results showed that I*GF1R*, *AKT1*, and *cyclin D2* showed an overall decreasing trend and were significantly lower than the negative control group after 48 h post-interference (*p* < 0.05). A gradual increasing trend was observed after 72 h. *MAPK1* exhibited an overall increasing trend and were significantly higher than the negative control group at both 48 h and 72 h (*p* < 0.05, Figure 7B, Appendix A).

### 3.5. Prokaryotic Expression and Purification of HcIGF1 Fusion Protein

The BL21(DE3) containing recombinant plasmid was induced by IPTG for 3 h at 25 °C or 37 °C or for 5 h at 15 °C, and the supernatant and precipitate were collected after ultrasonic fragmentation. The results of SDS-PAGE showed that there was a specific band at about 26 kD in both the supernatant and the precipitate after 3 h induction at 37 °C. The pET32a vector had a 6× His tag at the C-terminus and the theoretical molecular weight of the recombinant protein should be about 26.51 KD, which was consistent with the expected size of the target protein. There was no such band in the control group without IPTG induction (Figure 8A), indicating the successful fusion expression of the target protein.

As verified by SDS-PAGE, the specific protein bands were obvious in the supernatant, so the specific protein was mainly present in soluble form. The eluate of 50 mM imidazole was used for multiple elution. The SDS-PAGE analysis showed that the first portion of 50 mM imidazole expressed protein was successively eluted; after the second elution, the loss of target protein was small, but there were still some non-specifically bound impurity proteins. After the fourth elution, a single electrophoresis band was obtained, but there was a higher level of target protein loss (Figure 8B).

The purified HcIGF1 recombinant protein was detected by Western blotting using 6× His monoclonal antibody. The results showed that the purified recombinant protein, after hybridization with 6× His monoclonal antibody, showed obvious bands at about 26 kD, while the control group had no clear band, which confirmed that the purified recombinant protein was HcIGF1 recombinant protein (Figure 8C).

### 3.6. Identification of Biological Activity of Recombinantly Expressed Proteins

The results showed that compared with the control group (0 μg/mL), the HcIGF1 recombinant protein could effectively promote the proliferation activity of *H. cumingii* cells (*p <* 0.05). Proliferation activity increased initially, and then slowly decreased with increasing concentration. Specifically, the 7.5 μg/mL to 22.5 μg/mL supplemental groups were able to significantly promote the proliferation activity of the cells within 3–6 days (*p* < 0.05). The 12.5 μg/mL supplemental group had the highest proliferation activity. This group exhibited a significant difference in proliferation activity compared with other supplemental groups (*p* < 0.05) and was able to highly significantly promote proliferation activity compared to the control group from 3–6 days (*p* < 0.01). The concentration continued to increase, but the growth-promoting effect decreased. The 2.5 μg/mL supplemental group promoted the proliferation activity of cells from 1–4 days (*p* < 0.05), but the effect was not significant from 5–6 days (*p* > 0.05, Figure 9).

## 4. Discussion

In recent years, it has been found that IGF1 is not only involved in physiological processes such as growth and development, reproduction, and nutrient metabolism, but also has a crucial role in inhibiting apoptosis of many types of cells and promoting cell proliferation and differentiation [2,28]. In this study, the *IGF1* gene was identified and isolated from *H. cumingii*. Sequence analysis revealed that HcIGF1 has the IIGF structural domain specific to IGFs, without the IGF2___C structural domain specific to the IGF2, and without the transmembrane structural domain, presumably entering the cell by binding to IGF receptor protein [29]. Amino acid similarity comparison indicated that the sequence of IGF1 from *H. cumingii* was highly similar to that of other bivalves, suggesting that the biological function of HcIGF1 is similar to that of other bivalves. However, the low homology with other vertebrates also suggests that the biological function of HcIGF1 may be somewhat different from that of vertebrates. 

Quantitative results showed that the gene was expressed in all tissues, indicating that it was involved in the regulation of various physiological functions of *H. cumingii*, but the expression in each tissue was different, with the highest expression in the foot. Studies have shown that IGF1 is the main substance mediating the growth promotion of growth hormone and has a dual stimulatory effect on myogenic cells, promoting both their proliferation and differentiation, while enhancing the contractility of regenerated muscle [30,31]. This implies that HcIGF1 may be involved in regulating the growth of muscle in *H. cumingii*. Expression was lowest in the mantle tissue, which also suggests that mantle tissue, being a highly differentiated tissue, has a low proliferation and differentiation capacity [24], which is why mantle cells are difficult to culture in vitro.

The in situ hybridization results for the mantle tissue showed that the hybridization signals were mainly concentrated in the outer fold, middle fold, and ventral mantle, indicating that the outer mantle had higher activity than the inner mantle. At the same time, the outer mantle plays a major role in the biomineralization process and is associated with Ca^2+^ storage and CaCO_3_ formation [32]. Studies have shown that IGF1 is one of the most abundant growth factors deposited in the bone matrix, stimulating osteoblast proliferation, accelerating their differentiation and enhancing bone matrix production, and is essential for coupling matrix biosynthesis to sustained mineralization [7,33]. In this study, we found that the hybridization signal of the outer mantle was significantly stronger than that of the inner mantle, suggesting that IGF1 may also play a role in the biomineralization process, which still needs to be further explored. 

In mammals, IGF1 binds to IGF1R and initiates multiple downstream signal transduction pathways, mainly through the PI3K/AKT signaling pathway and MAPK signaling pathway to activate transcription factors and transduce extracellular signals to the nucleus, exerting its biological effect of promoting cell proliferation and inhibiting cell apoptosis [34,35]. It was reported that in the study of NO-induced apoptosis in primary *hippocampal* neuronal cells, IGF1 was found to prevent apoptosis through the PI3K/AKT pathway, and the expression of AKT-deficient forms almost completely blocked the effect of IGF1, suggesting that AKT is an essential signaling molecule in the inhibition of NO-induced neuronal apoptosis by IGF1 [36]. In the present study, knockdown of IGF1 resulted in a significant decrease in the expression levels of *AKT1* and *IGF1R*, which is consistent with the above findings, and it is hypothesized that IGF1 may bind to IGF1R and exert its biological functions mainly through the PI3K/AKT pathway in *H. cumingii*. Cyclin D is an important cell-cycle regulator, a rate-limiting factor that promotes the G1/S transition and plays an important role in the transition from G1 phase to S phase [37]. It has been shown that IGF1 stimulates DNA synthesis and increases the number of S-phase embryonic cortical precursor cells in vitro, and that the protein expression level of *cyclin D* increases rapidly after 4 h of IGF1 treatment. In addition, the PI3K/Akt pathway underlies IGF1 activity, and blocking this pathway prevents mitosis [38]. In the present study, the expression of *cyclin D2* also decreased after interference with *HcIGF1*, and it is speculated that IGF1 may increase the transcriptional activity of cyclin D2 mainly through the PI3K/AKT pathway, which in turn promotes cell proliferation. The MAPK signaling pathway can also regulate cell-cycle progression and promote cell proliferation. IGF1 has been reported to promote the proliferation and osteogenic differentiation of dental pulp stem cells through activation of the MAPK pathway [39]. However, in the present study, the expression of *MAPK1* increased instead after *HcIGF1* was disturbed, the mechanism of which is currently unknown. Therefore, it is necessary to further investigate the mechanism of IGF1’s role in *H. cumingii*.

The prokaryotic expression system has advantages of being simple and having low costs, short time requirements, and high expression yield, making it the most mature protein expression system [40]. pET32a (+) is used as a prokaryotic expression vector with six histidine residues at the carboxyl terminus that attach to the tail of the expressed protein while increasing the soluble expression of the recombinant protein [41]. The SDS-PAGE results showed that pET32a-HcIGF1 was expressed in DE3 in both supernatant and inclusion bodies, but mainly in the soluble supernatant form, which may be related to the characteristics of the protein and the vector to meet the requirements of subsequent purification. Three temperatures were also selected for induction, and the results showed that a higher percentage of solubility was induced at 3 h and 37 °C, probably because protein synthesis was faster at 37 °C and the protein had not yet folded to form inclusion bodies [42]. The protein mainly existed in soluble form, which better maintained the protein activity, and laid the foundation for the next protein activity analysis.

The results of this experiment showed that the HcIGF1 protein was effective in promoting cell proliferation at all concentrations within 3–4 days. Cao et al. showed that the human IGF1 recombinant protein could effectively promote the proliferation of human breast cancer cells at the minimum concentration of 12.5 µg/mL [43]. Compared with this, HcIGF1 recombinant protein had better biological activity. Meanwhile, there was a difference in the level of action of each concentration addition group. The pro-proliferative effect of HcIGF1 protein at a concentration of 2.5 μg/mL was not significant in the 5–6 day period, probably because the HcIGF1 recombinant protein added to the medium was depleted. The pro-proliferative effect was most significant at a concentration of 12.5 μg/mL and decreased when the concentration increased to 17.5 μg/mL and 22.5 μg/mL. This may be related to the fact that IGF1 has both growth-promoting and apoptosis-inducing effects on cells [44].

## 5. Conclusions

In summary, the full-length cDNA of the *IGF1* gene of *H. cumingii* was cloned in this study. The results confirmed that HcIGF1 is relatively evolutionarily conserved but has low homology with other species. Localization analysis of *HcIGF1* by qRT-PCR and in situ hybridization showed that *HcIGF1* expression was lowest in the mantle tissue, which was directly related to the weak proliferation ability of coat membrane cells in vitro. The stronger hybridization signal in the outer mantle relative to the inner mantle suggested that the cellular activity of the outer mantle might be stronger, and also suggested that HcIGF1 might be involved in the biomineralization process. The RNAi results indicated that *HcIGF1* might play a biological role in promoting cell proliferation and inhibiting apoptosis through the PI3K/AKT and MAPK pathways. Subsequently, the pET32a-HcIGF1 recombinant expression vector was constructed, and induced expression and protein purification were performed. The biological activity of recombinant HcIGF1 was studied by adding it to the culture medium, and the results showed that the recombinant protein had the best pro-proliferative activity at a concentration of 12.5 μg/mL. These findings provide a theoretical basis for improving the proliferation rate of shellfish mantle cells in in vitro culture, passaging culture and establishing cell lines, and also lay the foundation for exploring the study of IGF1 in the regulation of shellfish growth and development.

## Figures and Tables

**Figure 1 biology-11-01369-f001:**
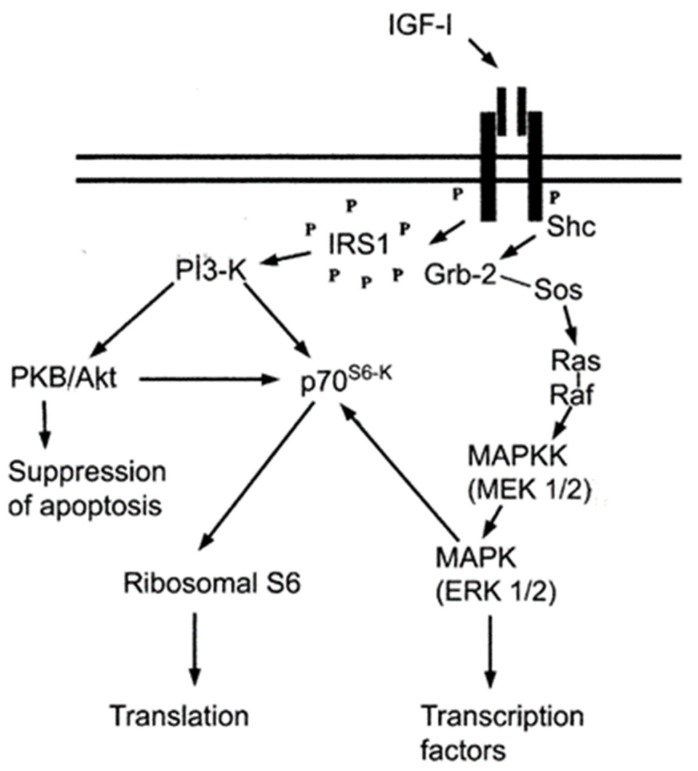
Diagram showing the intracellular signaling pathway of IGF1.

**Figure 2 biology-11-01369-f002:**
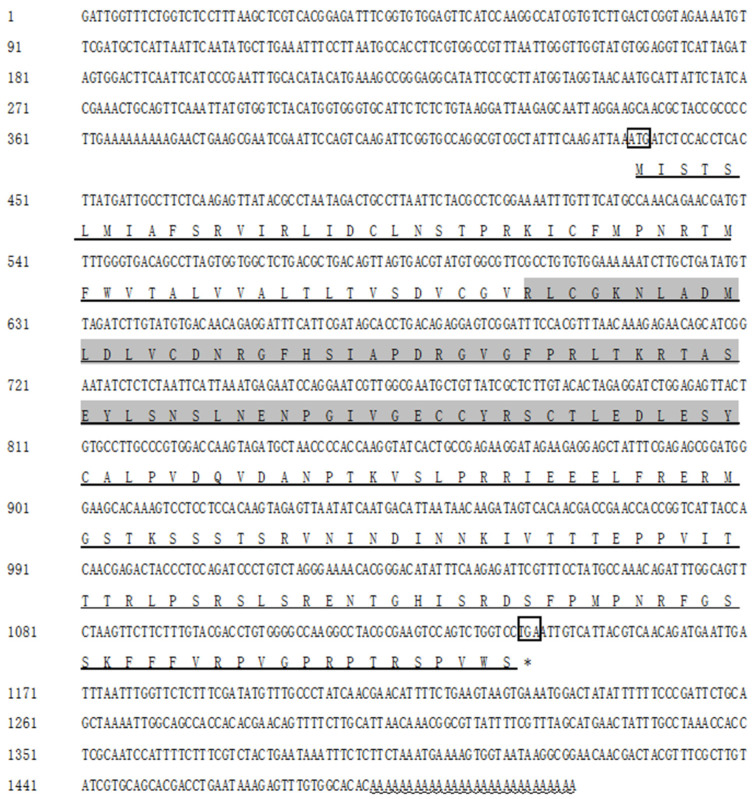
Nucleotide and deduced amino acid sequences of *HcIGF1*. Black boxes represent start and stop codons; gray boxes represent amino acids in the IIGF structural domain; underlined horizontal lines represent amino acids encoding the protein; underlined wavy lines represent the poly A structure.

**Figure 3 biology-11-01369-f003:**
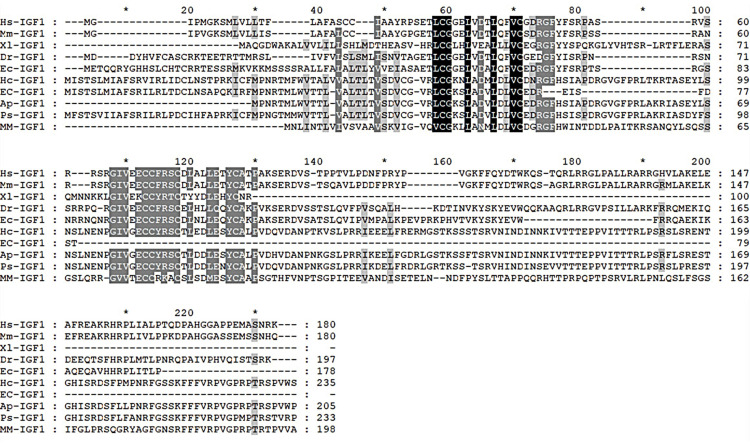
Sequence comparison of homology of IGF1 with other species with *H. cumingii.* Other species NCBI accession numbers: *Homo sapiens* (Hs): NP_000609.1; *Mus musculus* (Mm): NP_001300939.1; *Xenopus laevis* (Xl): NP_001156865.1; *Danio rerio* (Dr): NP_571900.1; *Epinephelus coioides* (Ec): AMR58932.1. *Elliptio complanate* (EC): GAHW01021065.1; *Amblema plicata* (Ap): GITL01141792.1; *Potamilus streckersoni* (Ps): GJAA01011606.1; *Mercenaria mercenaria* (MM): XP_045198864.1. Black: conserved amino acid residue; gray: analogous residues.

**Figure 4 biology-11-01369-f004:**
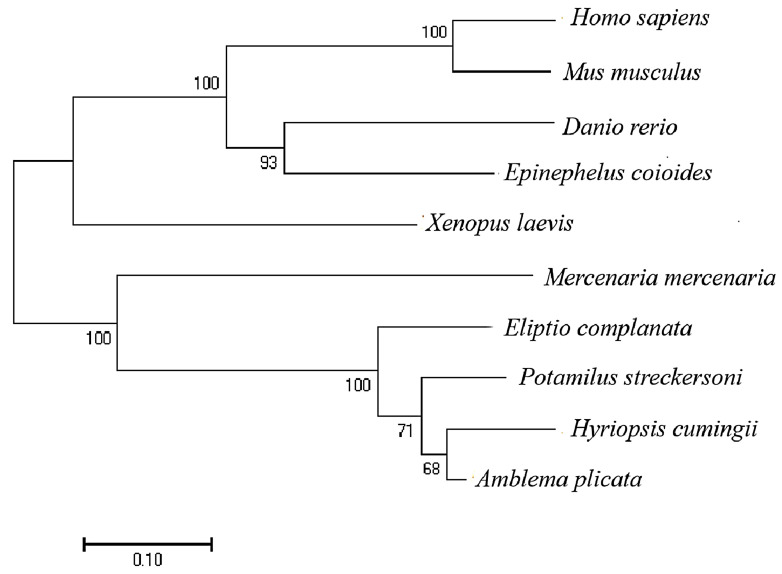
Phylogenetic analysis of IGF1 protein from different species. GenBank accession numbers are as in Figure 3, the number on the node indicates the confidence value of the test for 1000 bootstrap repetitions.

**Figure 5 biology-11-01369-f005:**
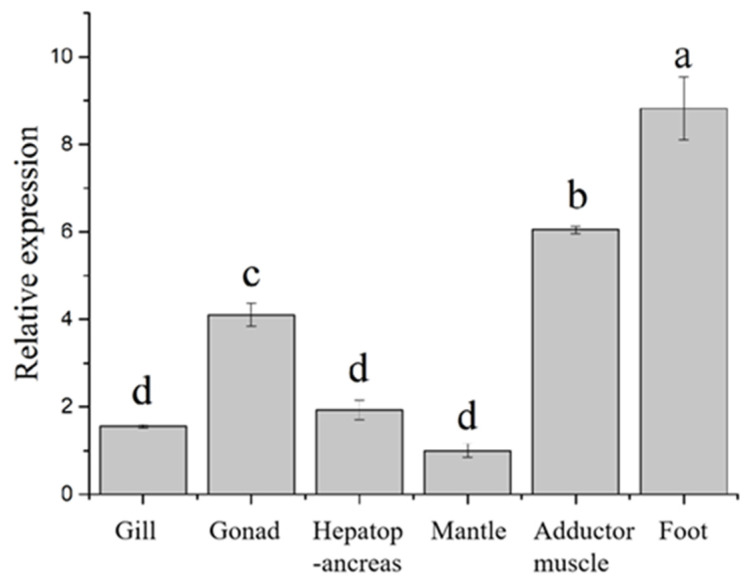
Expression of *HcIGF1* in different tissues. Different letters indicate significant differences (*p* < 0.05).

**Figure 6 biology-11-01369-f006:**
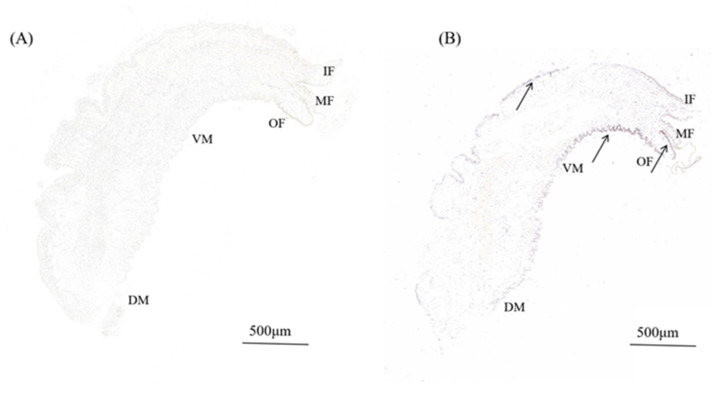
In situ hybridization in the mantle of *H. cumingii*. (**A**) Micrograph of the control groups. (**B**) Signal of HcIGF1 in the experimental groups. The arrow indicates the position of the signal. OF, outer fold; MF, middle fold; IF, inner fold; DM, dorsal mantle; VM, ventral mantle.

**Figure 7 biology-11-01369-f007:**
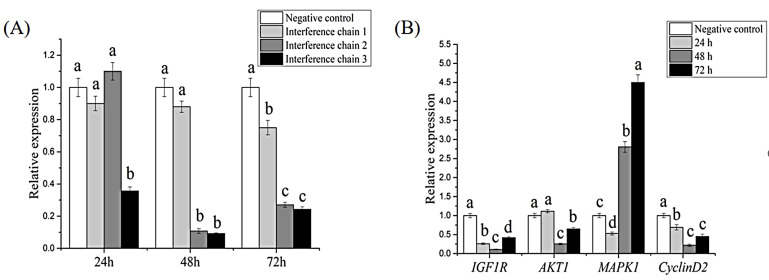
(**A**) Relative expression levels of *HcIGF1* in the mantle after interference. Interference chains 1, 2, and 3 represent G1, G2, and G3, respectively. Different letters at the same time point indicate significant differences (*p* < 0.05). (**B**) Relative expression of downstream genes of *HcIGF1* in the mantle after interference. Different letters for the same gene indicate significant differences (*p* < 0.05).

**Figure 8 biology-11-01369-f008:**
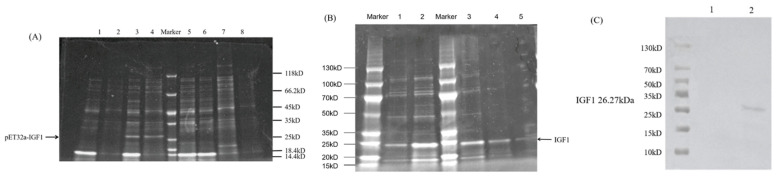
(**A**) Soluble analysis of HcIGF1 protein. 1: Supernatant of uninduced bacteria; 2: precipitation of uninduced bacteria; 3: supernatant after 3 h induction at 37 °C; 4: precipitation after 3 h induction at 37 °C; 5: supernatant after 3 h induction at 25 °C; 6: precipitation after 3 h induction at 25 °C; 7: supernatant after 5 h induction at 16 °C; 8: precipitation after 5 h induction at 16 °C. (**B**) SDS-PAGE analysis of purified HcIGF1 protein. 1, 2, 3, 4, and 5 represent the 1st, 2nd, 3rd, 4th, and 5th elution, respectively. (**C**) Western blot identification. 1: pET-32a (+) empty vector control; 2: IGF1 protein. Arrows mark the target bands.

**Figure 9 biology-11-01369-f009:**
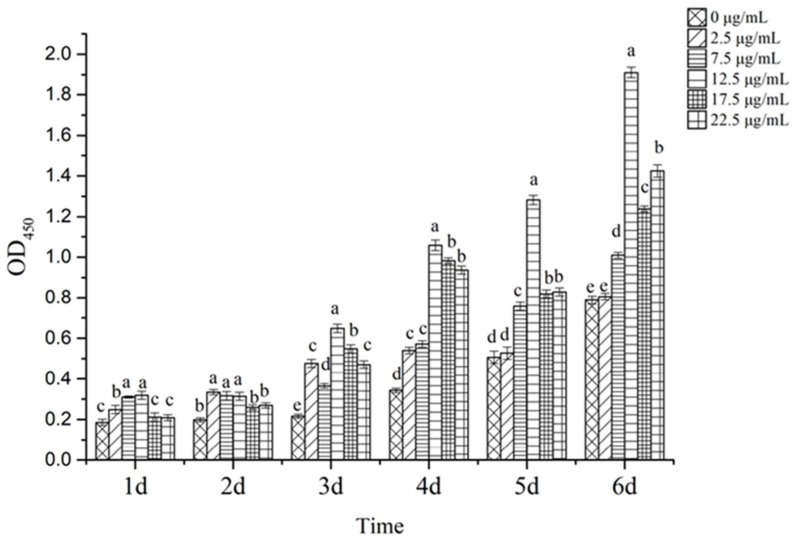
Effect of different concentrations of HcIGF1 protein on the growth of mussel mantle cells. Different letters for the same day count indicate significant differences (*p* < 0.05).

**Table 1 biology-11-01369-t001:** Primers used in the experiment.

Primer Name	Sequence (5′ to 3′)	Purpose
IGF1	F: TTTGGGTGACAGCCTTAG	partial sequence PCR
	R: TCCTTCTCGGCAGTGATA	
IGF1-5′	Outer: ACGGGCAAGGCACAGTAACTCTCCA	5′RACE
	Inner: TTTTTCCACACAGGCGAACGCCACA	
IGF1-3′	Outer: AGTGGTGGCTCTGACGCTGACAGTT	3′RACE
	Inner: TGCCTTGCCCGTGGACCAAGTAGAT	
qRT-PCR	F: AGAGTTACTGTGCCTTGCC	qRT-PCR
	R: ACTTTGTGCTTCCCATCC	
EF1α	F: AGGGTCCTTCAAGTATGCC	Reference gene
	R: CCAGTTTCCACTCTGCCTA	
IGF1R	F: ATTCCTCAATCGTCGCTCAA	Downstream Genes
	R: TGCGGAGTTCCAATGCTG	
cyclin D2	F: GATAGCAGCAGGGAGTGT	
	R: TTTGTGGATTCCGTTTCG	
MAPK1	F: AGGTTGGTCCTCGCTACT	
	R: TGGTGCCCGTATTATGTC	
AKT1	F: GTGAACAGAAGGCGAAAG	
	R: ATCCAGGGTCTCAGCATC	
G1	F: CAAGGTATCACTGCCGAGAA	DsRNA
	R: TGGATTGCGAGGTGGTTT	
T7-G1	F: GGATCCTAATACGACTCACTATAGGCAAGGTATCACTGCCGAGAA	
	R: GGATCCTAATACGACTCACTATAGGTGGATTGCGAGGTGGTTT	
G2	F: GTTATACGCCTAATAGACTGC	DsRNA
	R: ACTTTGTGCTTCCCATCC	
T7-G2	F: GGATCCTAATACGACTCACTATAGGGTTATACGCCTAATAGACTGC	
	R: GGATCCTAATACGACTCACTATAGGACTTTGTGCTTCCCATCC	
G3	F: GCCTTCTCAAGAGTTATACGC	DsRNA
	R: ACTTTGTGCTTCCCATCC	
T7-G3	F: GGATCCTAATACGACTCACTATAGGGCCTTCTCAAGAGTTATACGC	
	R: GGATCCTAATACGACTCACTATAGGACTTTGTGCTTCCCATCC	
IGF1-ISH	F: AAAGTCCTCCTCCACAAG	In situ hybridization probe
	R: GGATCCTAATACGACTCACTATACCACAGGTCGTACAAAGAA	
P-IGF1	F: TTGTCGACGGAGCTCGAATTCATGATCTCCACCTCACTTATGATTGC	Plasmid construction
	R: GCTGATATCGGATCCGAATTCTCAGGACCAGACTGGACTTCGC	

## Data Availability

All the data presented in this study are included in the article. If needed, Appendix A is available on request from the corresponding author.

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
