# Peer review of "Identification and Functional Analysis of the Cell Proliferation Regulator, Insulin-like Growth Factor 1 (IGF1) in Freshwater Pearl Mussel (Hyriopsis cumingii)"

_biology, 2022, doi:10.3390/biology11091369_

Round 1

Reviewer 2 Report

The manuscript describes a series of experiments to first discover the IGF1-like gene in freshwater pearl mussels, and then knock out the gene to identify which pathways the gene is involved. While there was a lot of effort put in the study and the results are useful for the scientific community, the materials and methods and the results sections were very difficult to understand and lack of information. In particular, the part describing the statistical analysis. 

Introduction

L61 – can you give the common name of species too.

L67-80. Very difficult to understand. What is apoptosis? You did no define it. Maybe you could have a figure that represent the pathways?

M&M

L113 – Did you check exon-exon boundaries

L119 – compared with known sequences: which species?

L128 – what are the physical parameters?

L129 – any input parameters for these programs to be able to re-do the analysis? Have you done any synteny analysis?

L133 – Should this paragraph be above the bioinformatics analysis?

L135 – there is no description of cycle time and temperature to be reproducible results

L138 – Why not using the PFaffl method, which is commonly adopted?

L139 – Why did you use only one house keeping gene?

L153 – I am very confused with the ordering of the paragraphs

L201 – What data did you analysed, what statistics did you and why are you trying to demonstrate here. The name of the programs that you use is quite irrelevant here, but what statical tests did you use is what is important, what model. You need to describe all of that. Multiple comparisons? I do not even know what are you comparing.

Results

Figure 2 and Figure 3 – you are using the latin name of species. Sometimes, it might be easier for the reader to also include the common name of the species as well.

Figure 5. Be careful with the colour as when I printed the manuscript the left picture of the figure was not even visible

Figure 6 – you never defined what is interference chain. You referenced Wang et al, but at this stage, because you did not describe any of you statistical analysis. What is interference chain 1, 2 and 3? What analysis does that refers too.

Figure 8 – Unreadable. Use different colours instead of different filling of the bars. What are the letter – where is Figure 8 mentioned?

Discussion

L224 to 342- Should it not be results?

Reviewer 3 Report

Suggested title: Identification and functional analysis of the cell proliferation regulator, insulin-like growth factor 1 (IGF1) in freshwater pearl mussel (Hyriopsis cumingii)

Statistical analysis: Multiple comparison should be performed by Tukey's HSD Test and results and discussion should be modified accordinlgly. DMRT is a very old method with less sensitivity.

Fig. 1: Underlined wave lines are not prominent, make them bolder for better identification.

Fig. 7 (B): The target protein lies below the 25 kD marker. How can the authors confirm that IGF1 is 26.27 kD?

Fig. 7 (C): It also looks below or near 25 kD marker. There is no arrow that indicates the target band.

Round 2

Reviewer 1 Report

Review (v2) in the attachement

Author Response

请参阅附件。

Reviewer 2 Report

While you have responded to the comment about the data analysis paragraph, you have not really changed it. I still don't what is the purpose of the statistical analysis undertaken and what data did you use as input.

Round 3

Reviewer 1 Report

Comments in attachment
